# Chaos and Predictability in Ionospheric Time Series

**DOI:** 10.3390/e25020368

**Published:** 2023-02-17

**Authors:** Massimo Materassi, Tommaso Alberti, Yenca Migoya-Orué, Sandro Maria Radicella, Giuseppe Consolini

**Affiliations:** 1Consiglio Nazionale delle Ricerche-Istituto dei Sistemi Complessi (CNR-ISC), Via Madonna del Piano 10, Sesto Fiorentino, 50019 Firenze, Italy; 2Istituto Nazionale di Geofisica e Vulcanologia, Via di Vigna Murata 605, 00143 Rome, Italy; 3The Abdus Salam International Centre for Theoretical Physics (ICTP), Strada Costiera 11, 34151 Trieste, Italy; 4Boston College, USA (BC), Institute of Scientific Research, Chesnut Hill, MA 02467, USA; 5National Institute of Astrophysics (INAF), Institute for Space Astrophysics and Planetology (IAPS), Tor Vergata, Via del Fosso del Cavaliere 100, 00133 Rome, Italy

**Keywords:** predictability, ionosphere, embedding phase space

## Abstract

Modelling the Earth’s ionosphere is a big challenge, due to the complexity of the system. Different first principle models have been developed over the last 50 years, based on ionospheric physics and chemistry, mostly controlled by Space Weather conditions. However, it is not understood in depth if the residual or mismodelled component of the ionosphere’s behaviour is predictable in principle as a simple dynamical system, or is conversely so chaotic to be practically stochastic. Working on an ionospheric quantity very popular in aeronomy, we here suggest data analysis techniques to deal with the question of how chaotic and how predictable the local ionosphere’s behaviour is. In particular, we calculate the correlation dimension $D_{2}$ and the Kolmogorov entropy rate $K_{2}$ for two one-year long time series of data of vertical total electron content (vTEC), collected on the top of the mid-latitude GNSS station of Matera (Italy), one for the year of Solar Maximum 2001 and one for the year of Solar Minimum 2008. The quantity $D_{2}$ is a proxy of the degree of chaos and dynamical complexity. $K_{2}$ measures the speed of destruction of the time-shifted self-mutual information of the signal, so that $K_{2}^{-1}$ is a sort of maximum time horizon for predictability. The analysis of the $D_{2}$ and $K_{2}$ for the vTEC time series allows to give a measure of chaos and predictability of the Earth’s ionosphere, expected to limit any claim of prediction capacity of any model. The results reported here are preliminary, and must be intended only to demonstrate how the application of the analysis of these quantities to the ionospheric variability is feasible, and with a reasonable output.

## 1. Introduction

The Earth’s ionosphere is a very rich open system [1] that interacts with the Solar Wind, the Earth’s magnetosphere, the neutral atmosphere including the troposphere, the cosmic radiation and, very likely, with the Earth’s lithosphere [2].

Due to its relevance for human activities, such as navigation and positioning, power plant and pipeline safety, great efforts have been done to make models of it, enabling predictions of its behaviour (see for instance the book [3], where some different authors review the first principle and empirical ionospheric models). The prediction of ionospheric behaviour, on a great variety of space and time scales, has made great progress in the history of aeronomy and space science since the “discovery of the ionosphere” by Marconi, Appleton, and Barnett in the 1920s. Still, it does make sense to raise the general question of to which extent this behaviour may really be predicted. Since the pioneering studies of Lorenz [4], physicists have realised that even *perfectly deterministic systems*, the dynamics of which may be written in closed form, show a certain degree of unpredictability, due to the phenomenon of *chaos*, whenever *non-linearity* comes into the play.

This is the case of the components of the Earth’s atmosphere as well [3,5]: in every possible model of a local or global portion of the ionosphere, any predicted quantity X is always expected to have some fluctuations δX, irregular and apparently out of reach for our prediction. These space and time irregularities represent an effect of the non-linear components of the ionospheric dynamics, working as a magnifying lens on the effects of the matter’s granularity, as it happens with fluid turbulence [6]. While making physical, empirical, machine-learning-based models of the ionosphere, one would not mind to know *if* the precision to predict some quantity may be reduced arbitrarily, or if any limit to this predictability exists. Such issues make perfect sense in the field of tropospheric weather prediction [4,5], so there should be no surprise for them to make sense in ionospheric weather and climate as well [7,8,9].

This paper is dedicated to presenting some data analysis tools, commonly used to assess the predictability of complex systems, which can be used to study the same aspect for the ionosphere. The application of the data analysis tools presented is suggested for the study of the *vertical total electron content* (vTEC), a physical quantity largely used to describe the local ionosphere, and of which huge worldwide continuously data monitoring exists. The definition of the vTEC on the top of a ground location of geographical coordinates $\left(\varphi ,\lambda \right)$ reads:(1)$$T_{\mathrm{vert}}^{\left(\varphi ,\lambda \right)}\left(t\right)=\int_{0}^{h_{\mathrm{GNSS}}}N_{e}\left(\varphi ,\lambda ,z,t\right)dz,$$
where $N_{e}$ is the free electron density number (in (Equation 1) *z* is the quote and *t* is the time). The dynamics of the density number $N_{e}$ of free electrons is very rich, and it should be expected to show all the characteristics defining “complexity” [5], precisely as it happens in meteorology. The complex dynamics of $N_{e}$ are necessarily reflected in a complex evolution of $T_{\mathrm{vert}}^{\left(\varphi ,\lambda \right)}$. The choice of vTEC as a proxy of the local ionosphere state makes practical sense: indeed, *the total electron content along a general path γ* is very useful in the field of ionospheric radio propagation, being proportional to the *optical path contribution* due to the ionospheric medium along *γ* [10]. The vTEC on the top of a certain ground location of coordinates $\left(\varphi ,\lambda \right)$ is meant to give an idea of the effect of the local ionospheric medium on radio propagation.

Ionospheric complexity may result in precise mathematical terms when a representation of the ionospheric medium is chosen. Consider, for instance, the *fluid representation of the ionospheric medium* (FRIM): the evolution of $N_{e}$ is described by a system of coupled, time dependent, and partial differential equations (PDEs) in which the density number, bulk velocity, and temperature of each chemical component of the ionosphere are involved as classical fields. Moreover, the geomagnetic and geoelectric field equations couple with those fluid PDEs. This would be by itself enough to expect complex dynamics to develop [3], namely *high dimensional chaos* [11]. Moreover, the FRIM is not even the most “detailed” representation possible: it is a complex, but still deterministic picture [3]; representations including *fluctuations* may be stochastic variations of the FRIM, such as the representation of the sporadic spread F layer in Ref. [12] or the kinetic pictures in Ref. [13].

The complexity of the dynamics just mentioned is expected to be reflected in the vTEC time series, as indeed it is. The evolution pattern of $T_{\mathrm{vert}}^{\left(\varphi ,\lambda \right)}\left(t\right)$ with time appears as *quasi-periodic*: the main component of this evolution is the *diurnal variability*, driven by insolation. Besides this periodicity, however, a huge variety of shapes appear in the $T_{\mathrm{vert}}^{\left(\varphi ,\lambda \right)}\left(t\right)$, all encoding the complexity of the ionospheric dynamics: this renders the evolution $T_{\mathrm{vert}}^{\left(\varphi ,\lambda \right)}\left(t\right)$ scarcely predictable. Assessing the limit of predictability of the vTEC, if any, a statement is made on the extent to which the ionospheric medium may be *predictably modelled*, i.e., represented deterministically [3].

In the present work, the data analysis techniques applied make use of concepts well known in the literature of *complex systems*, which become popular in the early 1990s in the field of magnetospheric physics (see, e.g., Refs. [14,15] and the later work of Ref. [16] and references therein), but less so in the field of ionospheric dynamics, although important attempts have been made in the past [17]. In particular, three concepts are used in our vTEC analysis: the concept of *embedding phase space*, that of *correlation dimension* $D_{2}$, and that of *Kolmogorov entropy*$K_{2}$ [18] (the symbols $D_{2}$ and $K_{2}$ refer directly to the way the correlation dimension and Kolmogorov entropy are calculated: see below).

The aforementioned quantities are well defined when one deals with an *autonomous finite dimensional* dynamical system Σ, described via its trajectories $X\left(t\right)$ throughout some phase space V, with finite dimension dimV=m. The dynamics
(2)$$\dot{X}=F\left(X\right)$$
of Σ determine the local properties of the trajectories: in particular, how “irregular” they are, “filling” a region of V as chaotic curves, which is described by the correlation dimension; and how fast the information shared by the present state $X\left(t\right)$ with the past ones $X\left(t^{\prime }<t\right)$ is lost, which is given by the Kolmogorov entropy rate.

The physical system in our case is the *local ionospheric medium* (LIM), of vTEC (Equation 1), where “local” means the correspondence of the given ground coordinates $\left(\varphi ,\lambda \right)$. As we have sketched before, the mathematical representations of the ionospheric medium are more complicated than a “simple” *m*-dimensional Σ: both the FRIM and all the possible kinetic representations of the “dirty plasma” [1,3] have an infinite dimensional V, as they are in practice field theories. In fact, one should think of the ionosphere as a fluid that may be in different conditions, ranging from “laminar” to “turbulent” flows: hence, it may show different behaviours, described via phase spaces of a different *finite* dimension, depending on how many physical modes are “switched on” in the flow conditions at hand. The use of finite dynamical systems in fluid dynamics is already well known in the literature, since Lorenz defined his paradigmatic 3-dimensional chaotic system to represent a simplified model for the atmospheric convection [4]: of course, the continuum mechanics of the atmosphere is an infinite-dimensional system as it would be the case for a kinetic representation of it. Yet, some selected modes of it, coupled among themselves but decoupled from, e.g., smaller scale ones, may well be described via a finite-dimensional Σ.

In order to obtain the information about the vTEC predictability, namely the predictability of the local ionospheric state, we consider the time series $T_{\mathrm{vert}}^{\left(\varphi ,\lambda \right)}\left(t\right)$ as *the only physical information available*, and look for a “suitable Σ” that can mimic the LIM physics. In particular, we apply the *embedding phase space analysis* obtained from the important works by Takens and others, see for instance Ref. [18] and the many references therein (in particular, Refs. [19,20]). The procedure, well known in the literature, and already applied by the Authors T.A. and G.C. to the Space Weather research [16], is worth briefly discussing in terms of how far the assumptions behind it will fit the LIM dynamics (for technical details the reader is addressed to the quoted references [18,19,20]).

The paper is organized as follows.

In Section 2 the dynamical system tools to be applied to the vTEC time series are introduced, and in Section 3 the outcome of their application to two vTEC time series is presented; one series pertains to a year of Solar Minimum and the other one to a year Solar Maximum: this choice is expected to make the analysis explore different helio-geophysical conditions, as the solar activity is the main trigger of the ionospheric response, called *Space Weather* [3].

Section 4 is finally devoted to the conclusions and physical reasonings regarding the presented results, and also some developments that are on their way.

## 2. Embedding Phase Space, $D_{2}$ and $K_{2}$

As stated in Section 1, we are trying to infer some dynamical information about the physical system “LIM around the location $\left(\varphi ,\lambda \right)$”, being able to work only with the time series $T_{\mathrm{vert}}^{\left(\varphi ,\lambda \right)}\left(t\right)$. This is done by constructing a finite-dimensional dynamical system, assumed to be governed by some dynamics as in Equation (Equation 2), solely out of the collection of values $\left\{T_{\mathrm{vert}}^{\left(\varphi ,\lambda \right)}\left(t\right)\right\}$. The tools presented here are able to give us the dimension
$$m=\dim V_{\mathrm{LIM}}$$
of the phase space $V_{\mathrm{LIM}}$, and other two quantities $D_{2}$ and $K_{2}$ that characterise the topological structure of the system trajectories $X\left(t\right)$ and their predictability. Nothing more will be inferred, instead, on the form of the function F giving the dynamics of X as in (Equation 2).

A description of the presented analysis tools is worth to be given.

### 2.1. The Embedding Phase Space V

Let us have a certain time series $\left\{y\left(t\right)\right\}$ as the only proxy collected for a given physical system. For $t\in \left[t_{i},t_{f}\right]$, *t* will assume only discrete values $t=t_{i}+n\tau_{s}$, being $\tau_{s}$ the sampling time and n=0,…,N−1, so that $t_{i}+\left(N-1\right)\tau_{s}=t_{f}$. The assumption is that the values $y\left(t\right)$ taken by the time series are due to the dynamics of a system Σ. In particular, if the system Σ is described by the finite-dimensional state $X=\left(X_{1},\ldots ,X_{m}\right)$, at each time the quantity *y* depends smoothly on all the components of X, such as $y\left(t\right)=g\left(X\left(t\right)\right)\;/\;g\in C^{\infty }\left(V,R\right)$. The aim of the data analysis tool described here is “to obtain Σ out of *y*″, i.e., to reconstruct the trajectory $X\left(t\right)$ out of the time series $y\left(t\right)$, for *t* in the interval of observation of *y*. Let us underline again that *this system* Σ *is completely unknown*, but some prior hypotheses on it are necessary:Along the interval $\left[t_{i},t_{f}\right]$, its physics is “stationary”, i.e., no sudden changes in the parameters or in the external forces take place, so that the dynamical structure in V remains the same. In practice, one assumes X to be governed by an *autonomous dynamics* as in (Equation 2), where F does not depend explicitly on time;The dimension m=dimV is unknown, and it will be an output of the embedding procedure (needless to say, this dimV must be *constant* along the time interval $\left[t_{i},t_{f}\right]$. As in our practical case, the condition of the ionospheric medium to range from hydrostatic equilibrium to bursty turbulence, *should be not* taken for granted. Reasonably, one should apply this technique to data sets of vTEC where the degree of turbulence of the LIM is constant, or accept to obtain results that are an average of all the different conditions of the LIM met throughout the time series. This latter condition is precisely the one met in the 1-year-long vTEC series analysis, as done here).

The result regarding whether finding a reasonable Σ out of $\left\{y\left(t\right)\right\}$ is possible, dates back to Takens’ Theorem [18], stating that an *m*-vector
(3)$$Y\left(t\right)=\left(y\left(t\right),z_{1}\left[y;t\right),\ldots ,z_{m-1}\left[y;t\right)\right),\;t\in \left[{\hat{t}}_{i},{\hat{t}}_{f}\right]\subseteq \left[t_{i},t_{f}\right],$$
formed by $y\left(t\right)$ and m−1 other quantities $z_{k}\left[y;t\right)$
*functionally depending* on $y\left(t\right)$, and *functionally independent* of each other, may work as a good state of Σ, existing as a smooth 1-to-1 relationship between Y and the “true state” X. In the definition (Equation 3) the brackets of $z_{k}\left[y;t\right)$ underline that, even if those quantities are *locally* dependent on *t*, they may depend *non locally* on $y\left(t\right)$, as it will be clear from the practical choice of our $z_{k}$s below. Note also that the system state is reconstructed in a possibly proper subinterval of $\left[t_{i},t_{f}\right]$, $\left[{\hat{t}}_{i},{\hat{t}}_{f}\right]$ with ${\hat{t}}_{i}\geq t_{i}$ and ${\hat{t}}_{f}\leq t_{f}$.

Once the vector $Y\left(t\right)$ is reconstructed, i.e., as we know the number *m*, and all the values $Y_{1,\ldots ,m}\left(t\right)$ for $t\in \left[{\hat{t}}_{i},{\hat{t}}_{f}\right]$, the topological properties of $X:\left[{\hat{t}}_{i},{\hat{t}}_{f}\right]\mapsto V$, characterizing the degree of chaos and predictability of Σ, are known, because one holds the curve $Y:\left[{\hat{t}}_{i},{\hat{t}}_{f}\right]\mapsto V$ that is in 1-to-1 infinitely differentiable correspondence to $X:\left[{\hat{t}}_{i},{\hat{t}}_{f}\right]\mapsto V$ (X and Y are said to be *diffeomorphic* to each other).

The chosen form for $Y\left(t\right)$ in (Equation 3) is
(4)$$Y\left(t\right)=\left(y\left(t\right),y\left(t+\Delta \right),y\left(t+2\Delta \right),\ldots ,y\left(t+\left(m-1\right)\Delta \right)\right):$$
the time Δ is *a time lag suitably chosen*, so that $y\left(t+k\Delta \right)$ is *functionally independent of its neighbours* $y\left(t+\left(k-1\right)\Delta \right)$ and $y\left(t+\left(k+1\right)\Delta \right)$. In practice, on the one hand, Δ must be *so large* that indeed the neighbouring $Y_{k}\left(t\right)=y\left(t+\left(k-1\right)\Delta \right)$ are independent; on the other hand, Δ cannot be *too large* because, after all, the components of $Y\left(t\right)$ in (Equation 4) must all refer “to the same state” of Σ; namely, the non-linear dynamics moving Σ has not to had the time to change X during a time interval of length Δ. In our analysis, Δ is chosen as the delay after which the mutual information between $y\left(t\right)$ and $y\left(t\pm \Delta \right)$ goes to less than 0.4bit (this results into approximately Δ=1200s for the data of vTEC at hand, see below). Clearly, other choices for Δ might be done: the one adopted here guarantees the $z_{k}\left[y;t\right)$ functions as in (Equation 3), realised as the $y\left(t+n\Delta \right)$ in (Equation 4), to be *functionally* independent of each other, and not only linearly independent; moreover, the mutual information is not reduced too much, so to render $y\left(t\right)$ and $y\left(t+q\Delta \right)$, with q=1,2,…,m−1, is still suitable to describe an *instantaneous state* $Y\left(t\right)$.

As the Δ is established, next one needs to fix *m* to construct $Y\left(t\right)$. In the present work, the right embedding dimension is, in a sense, established by looking at what would be the result of guessing it. The guess-and-refine work to obtain the proper m=dimV is done through the guess calculation of the quantity $D_{2}$: various guess dimV are tried, for each of them a guess $D_{2}$ is calculated, and the right dimV is chosen as the one giving the correct $D_{2}$.

### 2.2. How Chaotic? Defining $D_{2}$ and Fixing dimV

The dimension $D_{2}$ is intended (and calculated) as *the Hausdorff dimension* of the points in V lying along the curve $Y\left(t\right)$, and it is reasonable to state that the right *m* is the smallest one for which $D_{2}$ reaches a saturation value, as stated in a while. The fact that one calculates this $D_{2}$ means that the trajectory $Y\left(t\right)$ is expected to develop on an attractor A⊂V that has *real dimension* $\dim A=D_{2}$; there is *an implicit assumption that the dynamics has a certain amount of chaos*, otherwise A should have dimension 1. So, the quantity $D_{2}$ rather makes sense for irregular evolutions $Y\left(t\right)$, giving irregular time series $y\left(t\right)$. This is indeed the case of the vTEC time series analysed here, as one can see by looking at the plots in Figure 1 below. Considering $\mu \left(Y_{0},r\right)$ the amount of points of the trajectory $Y\left(t\right)$ around the point $Y_{0}\in V$ within a (small) neighbourhood of size *r*, the Hausdorff dimension $D_{2}$ is defined so that:(5)$$\mu \left(Y_{0},r\right)\propto r^{D_{2}\left(Y_{0}\right)}.$$In (Equation 5) the dimension $D_{2}$ may depend on the point around that is calculated (multi-fractal, or locally fractal, attractors), but in this work we are looking for a unique $D_{2}$ throughout the whole evolution studied, as $t\in \left[{\hat{t}}_{i},{\hat{t}}_{f}\right]$. In general, $D_{2}\leq m$: *the larger* $D_{2}$, the thicker is the attractor in V, through which the system trajectory evolves, i.e., *the more chaotic* its dynamics turns out to be. The value $D_{2}=1$ is that of an infinitely regular (smooth) curve, a value $1<D_{2}<m$ is chaos, while $D_{2}=m$ represents a *fully stochastic* evolution, i.e., *pure noise* (the idea of smoothness out of the necessarily discrete map of any real data is rather loose: on the one hand, it does not make sense to speak about any $C^{\infty }$ space of discrete time evolutions; on the other hand, the solution of a system as the $\dot{X}=F\left(X\right)$ thought of to govern Σ *is* infinitely derivable, as F is. In practice, the calculation of $D_{2}$, giving rise to some number in between 1 and dimV, with some uncertainty of course, provides us with an idea of how chaotic the dynamics of Σ is. Moreover, “noise” does not mean “white noise” or “Gaussian noise”, rather *fully probabilistic evolution*: turbulence has made us used to probability distributions that represent all but “trivial” noise). A regular evolution with $D_{2}=1$ may be fully predicted, while the more chaotic it is, the less predictable it turns out to be. Fully stochastic evolutions with $D_{2}\to m$ should be treated only via probability.

Coming back to the evaluation of *m*, as a candidate embedding dimension *d* is given, it is possible to calculate the first quantity we are interested in, i.e., the correlation dimension $D_{2}\left(d\right)$, out of the curve $Y_{d}\left(t\right)=\left(y\left(t\right),y\left(t+\Delta \right),\ldots ,y\left(t+\left(d-1\right)\Delta \right)\right)$: due to its meaning, one has $D_{2}\left(d\right)\leq d$, and it can just grow with *d*, that is $D_{2}\left(d+1\right)\geq D_{2}\left(d\right)$. Hence, the procedure of attempting to use $V_{d}$ with larger and larger *d* stops when the correlation dimension $D_{2}$ stops growing as the embedding dimension is increased, that is for $d=\tilde{d}$ with $D_{2}\left(\tilde{d}+1\right)=D_{2}\left(\tilde{d}\right)$. Then, the embedding dimension *m* is chosen as $m=\tilde{d}$.

### 2.3. How Unpredictable? Defining $K_{2}$

The other quantity that will be calculated for our system is the Kolmogorov entropy $K_{2}$, representing, in practice, *the amount of trajectory location precision that is lost in a single time step* of the evolution (this $K_{2}$ calculated from $Y_{d}\left(t\right)$, depends on the candidate embedding dimension *d*, i.e., one expects to have $K_{2}\left(d\right)$). This quantity is calculated by considering ε-coarse graining of the reconstructed phase space V, so that all along the trajectory points are collected in *finite size neighbourhoods*, as in Figure 2. By simply point counting, one may calculate the joint probability measure $P\left(Y_{1},\ldots ,Y_{n}\right)$ that the system state is in the neighbourhood $U\left(Y_{1},\varepsilon \right)\overset{\mathrm{def}}{=}i_{1}$ at time $t_{1}$, in $U\left(Y_{2},\varepsilon \right)\overset{\mathrm{def}}{=}i_{2}$ at time $t_{2}>t_{1}$ and so on: the total Shannon uncertainty about the trajectory location after *n* times is defined as:(6)$$K_{n}=-\sum_{i_{0},\ldots ,i_{n}}P\left(Y_{1},\ldots ,Y_{n}\right)\log P\left(Y_{1},\ldots ,Y_{n}\right).$$With those $K_{n}$s, one may define the limit
(7)$$K_{2}=\lim_{\tau_{s}\to 0}\lim_{\varepsilon \to 0}\lim_{N\to +\infty }\frac{1}{N\tau_{s}}\sum_{n=0}^{N-1}\left(K_{n+1}-K_{n}\right),$$
that is our Kolmogorov entropy (rate), where *N* is the number of times considered. In (Equation 7) the limits $\lim_{\tau_{s}\to 0}$, $\lim_{\varepsilon \to 0}$ and $\lim_{N\to +\infty }$ rather indicate that the sampling time should be much smaller than the dynamics timescales, so that the neighbourhood size should be much smaller than the Y gross variability, and that enough data must be collected, respectively.

Some remarks are necessary for the definition (Equation 6) and its use (Equation 7). First of all, the trajectory $Y\left(t\right)$, reconstructed via the embedding procedure discussed above, *is namely fully known*, so that *a rigore* it should be non-sense speaking about an ignorance entropy. However, introducing the ε-graining, the resolution of our observation becomes finite, and some uncertainty must be admitted. This uncertainty is not a mere artefact of some entropy-fanatic: the quantity $K_{2}$ defined in (Equation 7) is in fact different for different systems, and is larger for more chaotic systems, i.e., systems with a higher degree of chaos diagnosed via other proxies, for instance Lyapunov exponents. This stated, one accepts that $K_{2}$, which is measured in bit/s, and is indeed an information entropy rate, is in practice the inverse of the time after which the ignorance about the system position increases 1 bit: ${\left(K_{2}\right)}^{-1}$ is *the timescale within which the behaviour of the system can be accurately predicted*.

Regarding the practical interpretation of $K_{2}$, a fully predictable evolution would show $K_{2}=0$, because it would be *predictable forever* as $\frac{1}{K_{2}}\to +\infty$; a chaotic system has a finite $K_{2}>0$, that is *predictability within some time* $\frac{1}{K_{2}}>0$; last but not least, a fully stochastic system shows $K_{2}\to +\infty$, so that the predictability horizon would be *a zero time* $\frac{1}{K_{2}}=0$, and the evolution is somehow a continuous dice rolling.

### 2.4. Practical Calculation of $D_{2}$ and $K_{2}$

In the data analysis performed here, the correlation dimension $D_{2}$ and the Kolmogorov entropy $K_{2}$ are calculated through the numerical recipes by Grassberger and Procaccia [21], whose work rendered those abstract quantities more easily calculable in practice [16]. First of all, one defines a *correlation intergral* $C_{m}\left[Y,r\right)$ as
(8)$$C_{m}\left[Y,r\right)=\lim_{N\to +\infty }\frac{1}{N^{2}}\sum_{i,j=1}^{N}\Theta \left(r-∥Y_{i}-Y_{j}∥\right).$$In (Equation 8), Θ is the Heaviside step function, while the symbol $∥Y_{i}-Y_{j}∥$ is the distance in $V_{m}$ between the two points $Y_{i}$ and $Y_{j}$ along the trajectory embedded in a $V_{m}$ space of dimension *m*: usually this is calculated as an Euclidean norm, or the *m*-dimensional Pitagora Theorem $∥Y_{i}-Y_{j}∥\overset{\mathrm{def}}{=}\sqrt{\sum_{h=1}^{m}{\left({\left(Y_{i}\right)}_{h}-{\left(Y_{j}\right)}_{h}\right)}^{2}}$; instead, r>0 is simply a real positive number. Then, one assumes this integral $C_{m}\left[Y,r\right)$ to be a *power law* in the limit of small *r*:$$\lim_{r\to 0}C_{m}\left[Y,r\right)=O\left(r^{D_{2}}\right).$$

Starting from this correlation integral, the correlation dimension $D_{2}$ is finally calculated as:(9)$$D_{2}=\lim_{r\to 0}\frac{\log C_{m}\left[Y,r\right)}{\log r}.$$

Once the quantity in (Equation 9) is calculated for the time series $Y\left(t\right)$ of interest, i.e., for the original $y\left(t\right)$ of real data, the degree of “chaoticity” of the dynamics giving rise to $y\left(t\right)$ may be assessed as stated before.

The Grassberger–Procaccia method described in Ref. [21] allows us to calculate $K_{2}$ practically: a satisfactory approximation of the quantity defined in (Equation 7) is indeed obtained by using the correlation integrals in the *m* and m−1 embedding dimensions as
(10)$$K_{2}=\lim_{r\to 0}\frac{1}{\tau_{s}}\log\left(\frac{C_{m-1}\left[Y,r\right)}{C_{m}\left[Y,r\right)}\right),$$
where $\tau_{s}$ is still the sampling time. Equation (Equation 10) is a powerful estimation of $K_{2}$ and it directly comes from inserting Equation (Equation 8) into Equation (Equation 7), since, as shown by Grassberger and Procaccia [21],
(11)$$C_{m}\left[Y,r\right)∼r^{D_{2}}\;e^{-\tau_{s}K_{2}}.$$

With the two operative formulas (Equation 9) and (Equation 10), we are now in the position of applying these tools to the evolution of the vTEC.

## 3. Application to the vTEC Time Series: Results

The discussion developed in Section 2 may now be applied to the study of the vTEC dynamics. In particular, we are going to study the case of the *mid-latitude ionosphere* by applying the aforementioned techniques to two time series $T_{\mathrm{vert}}^{\left(\varphi ,\lambda \right)}\left(t\right)$, representing respectively the vTEC on the top of the GNSS station of Matera during the Solar Maximum year 2001 and the Solar Minimum year 2008. The vTEC data are taken from the “ICTP Calibrated GNSS TEC Service” (https://arplsrv.ictp.it/, accessed on 12 February 2023) that makes use of the TEC calibration technique described in Ref. [22].

The station of Matera has the following *geographic coordinates* [23]
$$\left(\varphi_{0},\lambda_{0}\right)=\left(40.649{}^{\circ },16.704{}^{\circ }\right),$$
and is generally indicated with the acronym MATE00ITA. In the (heavy) symbols of Equation (Equation 1), the two time series will hence be indicated as $T_{\mathrm{vert}}^{\left(\varphi_{0},\lambda_{0}\right)}\left(2001,\mathrm{DOY}\right)$ and $T_{\mathrm{vert}}^{\left(\varphi_{0},\lambda_{0}\right)}\left(2008,\mathrm{DOY}\right)$, where “DOY” is the day-of-the-year. The vTEC are expressed in TEC units (1 TECu= $10^{16}/m^{2}$ electrons) [5,10]. The time *t*, represented by the year and the DOY, has a resolution $\tau_{s}=30\;s$. The two vTEC time series will be more simply indicated as $y_{2001}\left(t\right)=T_{\mathrm{vert}}^{\left(\varphi_{0},\lambda_{0}\right)}\left(2001,\mathrm{DOY}\right)$ and $y_{2008}=\left(t\right)T_{\mathrm{vert}}^{\left(\varphi_{0},\lambda_{0}\right)}\left(2008,\mathrm{DOY}\right)$: this emphasizes how the vTEC time series plays the role of the physical proxy $y\left(t\right)$ discussed in Section 2.

The time series $y_{2001}\left(t\right)$ and $y_{2008}\left(t\right)$ are plotted in Figure 1: at a glance, one can see that the two evolutions show a similar seasonal time structure, but that of Solar Maximum $y_{2001}\left(t\right)$ has much wider excursions, due to the “higher” solar activity. Equally, one can note that in the year-long time series, geomagnetically and seasonally different conditions of the ionosphere are included, a fact that will have to be further discussed more ahead in this Section and in Section 4 below.

### 3.1. The M=3 Embedding and the Dimension $D_{2}$ for the vTEC

As a first step of the vTEC embedding analysis, we have determined the time lag Δ necessary to construct the vectors
$$\left\{\begin{array}{cc} & Y_{2001}\left(t\right)=\left(y_{2001}\left(t\right),y_{2001}\left(t+\Delta \right),\ldots ,y_{2001}\left(t+\left(m-1\right)\Delta \right)\right),\\ & Y_{2008}\left(t\right)=\left(y_{2008}\left(t\right),y_{2008}\left(t+\Delta \right),\ldots ,y_{2008}\left(t+\left(m-1\right)\Delta \right)\right). \end{array}\right.$$The value $\Delta =40^{\prime }$ has been chosen, as it is the one reducing the Δ-delayed self-mutual information of the $y\left(t\right)$ to less than 0.4bit; see Figure 3. The proper embedding procedure is performed as described in Section 2. As a guess, dimV is chosen, the correlation dimension $D_{2}$ of the evolution is calculated for the resulting trajectory in the phase space V: the larger the “tentative” dimV is, the larger the dimension $D_{2}$ turns out to be. The right m=dimV is the smallest one, for which $D_{2}$ reaches its constant saturation value. For both the time series $y_{2001}\left(t\right)$ and $y_{2008}\left(t\right)$, the embedding dimension found is:(12)m=3.Hence, the two quantities of interest in (Equation 9) and (Equation 10) read, respectively, as:(13)$$D_{2}=\lim_{r\to 0}\frac{\log C_{3}\left[Y,r\right)}{\log r},\;\;K_{2}=\lim_{r\to 0}\frac{1}{\tau_{s}}\log\left(\frac{C_{2}\left[Y,r\right)}{C_{3}\left[Y,r\right)}\right).$$

The fact that (Equation 12) holds both for the Solar Maximum time series $y_{2001}\left(t\right)$ and the Solar Minimum $y_{2008}\left(t\right)$ is already an interesting piece of information: it suggests that *in the mid-latitude*, the higher variability of the ionosphere during a Solar Maximum year is not big enough to render it necessarily a phase space V larger than in the Solar Minimum year, because no more degrees of freedom enter the play of dynamics, at least on average over a time of one year. The dynamical similarity between $y_{2001}\left(t\right)$ and $y_{2008}\left(t\right)$ may also be inspected by looking at a two-dimensional projection of the trajectories $Y_{2001}\left(t\right)$ and $Y_{2008}\left(t\right)$: this may be qualitatively understood by plotting $y\left(t+\Delta \right)$ against $y\left(t\right)$ for the two time series, obtaining the plots of Figure 4. Such plots should be the “shadow”, along one of the three possible coordinate planes in the phase space $V_{3}$ of the attractor A described by $Y\left(t\right)$. By a loose inspection of plots in Figure 4, one may see that, even if the range of the vTEC in the two cases is different, the “shape” of the attractor does not look very different for $y_{2001}\left(t\right)$ and $y_{2008}\left(t\right)$, essentially suggesting similar vTEC dynamics for the two years.

This similarity is *confirmed in numbers* when the $D_{2}$ is calculated for the two time series. Indeed, one obtains the values:(14)$$D_{2}\left(2001\right)≃2.78,\;D_{2}\left(2008\right)≃2.78,$$
i.e., *the same number* for the two time series $y_{2001}\left(t\right)$ and $y_{2008}\left(t\right)$. In Figure 5, $\log C_{3}\left[Y,r\right)$ is plotted against logr for the two years of vTEC: the two curves are well approximated by straight lines for r→0, the angular coefficient of which gives $D_{2}$ (please note that, for simplicity, only the dependence on *r* is highlighted in the symbol $C_{3}\left(r\right)$ in the plot); the two lines are parallel, i.e., they have the same slope, and hence $D_{2}\left(2001\right)=D_{2}\left(2008\right)$.

The result of $D_{2}\left(2001\right)$ and $D_{2}\left(2008\right)$ states a first assessment and raises a question about the vTEC dynamics. First of all, the dimA diagnosed by the results (Equation 14) is very high, considering dimV=3: this means that the $y_{2001}\left(t\right)$ and $y_{2008}\left(t\right)$ evolutions come from *highly chaotic dynamics*, which are *almost phase-space-filling*, i.e., *almost probabilistic*. This cannot directly mean that the ionospheric deterministic modelization is hopeless, but rather that one must do some work to disentangle the regular part of the vTEC evolution out of the whole signal, restricting the realm of “noisy dynamics”. Theoretically speaking, this will correspond to disentanglement in the ionspheric proxy $T_{\mathrm{vert}}^{\left(\varphi ,\lambda \right)}\left(t\right)$, the effect of the regular component $N_{0}$ from that of fluctuations δN in a decomposition $N_{e}=N_{0}+\delta N$ of the free electron number density (this subdivision is richly explained in Refs. [3,12] and references therein).

The question raised by the presented result (Equation 14) is that these two correlation dimensions have *surprisingly equal values* for the two years of Solar Maximum and Solar Minimum, and this may look like an apparent contradiction with the fact that *the helio-geophysical conditions of a year of Solar Maximum are different from those of Solar Minimum*: in the Solar Maximum period, the geomagnetic storms are more frequent, and hence more irregularity is expected in the series $y_{2001}\left(t\right)$. We will thoroughly comment on this question in Section 4 below, even though we have already underlined how the analysis covers the 1 year long series, and gives the necessary yearly average results.

### 3.2. The Komogorov Entropy $K_{2}$ for vTEC

Let us now present the findings for what $K_{2}$ is concerned. The Grassberger and Procaccia formulation, which leads to the recipe (Equation 10), is applied as in the $K_{2}$ expression in (Equation 13) to the time series data. In particular, considering the correct value of $m\overset{\mathrm{def}}{=}\dim V$ to be the one after which the dynamical properties of the reconstructed evolution $Y\left(t\right)$ out of $y\left(t\right)$ become stable, it is believed to be safer to use the ratio $\frac{C_{m}\left(r\right)}{C_{m+1}\left(r\right)}$ instead of $\frac{C_{m-1}\left(r\right)}{C_{m}\left(r\right)}$ in (Equation 10): so, we have used $\frac{C_{3}\left(r\right)}{C_{4}\left(r\right)}$ instead of the $\frac{C_{2}\left(r\right)}{C_{3}\left(r\right)}$ in (Equation 13). Considering the dependence of this $\frac{C_{3}\left(r\right)}{C_{4}\left(r\right)}$ on *r* for the two time series, it is possible to evaluate the value of $K_{2}$ as the logarithm of the intercept $\lim_{r\to 0}\frac{C_{3}\left(r\right)}{C_{4}\left(r\right)}$ of the plot with the vertical axis, divided by $\tau_{s}$; these plots are reported in Figure 6. The inverse $K_{2}^{-1}$ of the Kolmogorov entropy rate has been stated to represent the time after which the uncertainty on the trajectory location $Y\left(t\right)\in V$ has grown at least 1bit, i.e., the *predictability horizon*. The values calculated for the vTEC time series $y_{2001}\left(t\right)$ and $y_{2008}\left(t\right)$ read:(15)$$K_{2}^{-1}\left(2001\right)\cdot \mathrm{bit}≃5.32^{\prime },\;K_{2}^{-1}\left(2008\right)\cdot \mathrm{bit}≃8.49^{\prime }.$$Regarding the results (Equation 15), one may comment that the trajectory location ignorance growth appears to be slower in the year of Solar Minimum, i.e., the variability of the vTEC seems to be more predictable in 2008, as its predictability time horizon $K_{2}^{-1}$ is longer. All in all, apparently, the stronger influence of Solar Wind sudden impacts at Solar Maximum renders the local ionospheric evolution less predictable, again on a yearly average basis.

The values in (Equation 15) appear to suggest that the vTEC is predictable *within a time of some minutes*, slightly shorter in the year of Solar Maximum, but of the same order of magnitude. Despite this, one has in mind that the local vTEC evolution does have quasi-periodic patterns determined by the main source of ionization, i.e., insolation [10]: from day to day, one can still be sure of the local time interval during which the vTEC shows a minimum and a maximum, a growth and a decrease. Moreover, the gross behaviour of the vTEC with seasons is also rather well known. The daily “recurrence” of the vTEC behaviour with local time is clearly shown when the plot of $y\left(t\right)$ is zoomed in, for instance as done in Figure 7. In that figure, six days of vTEC of the year 2001 are shown, with a time resolution of 1 h for legibility: the periodic pattern due to the sunlight drive is clearly evident. Such a quasi-periodicity appears to invoke a predictability of 1 day, *at least in the shape of the regular part* of the $T_{\mathrm{vert}}^{\left(\varphi_{0},\lambda_{0}\right)}\left(t\right)$. This apparent contradiction between the short values of $K_{2}^{-1}$ in (Equation 15) and the 24 h-quasi-periodicity of the vTEC will be discussed in Section 4.

## 4. Conclusions, Issues, and Future Research

The ionospheric medium undergoes different mathematical representations as a dynamical system, ranging from the fluid mechanical FRIM to kinetic theories. In this paper, a data-driven approach is used to sketch a geometrical representation of the local ionospheric evolution as a *finite dimensional dynamical system*. In particular, we have constructed a finite dimensional phase space V through which the state of the local ionosphere X moves: this was done by reconstructing an equivalent state evolution $Y\left(t\right)$ via the *embedding procedure*, starting from a 1 year long vTEC time series $y\left(t\right)$, with 30s resolution. To give some more physical taste to this construction, and to confront it against what we expect from a gross knowledge of the geospace physics, two years of vTEC data were analysed, on the top of the same mid-latitude GNSS station Matera: the Solar Maximum year 2001 and the Solar Minimum year 2008, working on the series $y_{2001}\left(t\right)\mapsto Y_{2001}\left(t\right)$ and $y_{2008}\left(t\right)\mapsto Y_{2008}\left(t\right)$.

The embedding dimensions of the two phase spaces $V_{2001}$ and $V_{2008}$ are rather small, and equal: $\dim V_{2001}=\dim V_{2008}=3$, yet enough to host “chaotic” trajectories. As commented before, it is already significant that the local ionospheric medium shows the same dimension, for its phase space, in the Solar Maximum and Minimum year: in principle, this suggests that the vTEC dynamics are produced by *a physical autonomous system (Equation 2) of three dynamical variables*. Of course, the practical construction of those three quantities $X_{1}$, $X_{2}$, and $X_{3}$, e.g., as functions of the quantities describing the FRIM or kinetic theories, is all a different problem, and is definitely not dealt with here. Yet, our analysis ensures that the helio-geophysical conditions of the years 2001 and 2008 reduce the infinite-dimensional functional spaces of the FRIM, or of the kinetic theories, to some effective V
*de facto* similar to $R^{3}$.

In order to measure *how chaotic and how predictable* the two evolutions $Y_{2001}\left(t\right)$ and $Y_{2008}\left(t\right)$ are, their *correlation dimension* $D_{2}$ and the *Kolmogorov entropy rate* $K_{2}$ have been computed: $D_{2}$ is a measure of the Hausdorff dimension of the attractor A⊂V containing the trajectory $Y\left(t\right)$ (treated as a probabilistic evolution with ergodic motion within A); $K_{2}^{-1}$ is the time after which the uncertainty on $Y\left(t\right)$ grows at least 1bit, becoming *strictly* unpredictable.

The remarkable result is that there are no sensible differences between the Solar Minimum and Solar Maximum yearly vTEC evolutions, either in the m=dimV or in the Hausdorff dimension of the attractor $D_{2}=\dim A$. As announced in Section 3, this is a little bit surprising, more for dimA than for dimV. During a year of Solar Maximum, many more magnetic storms take place, and we expect that the three variables X describe topologically different trajectories in their V during *geomagnetically quiet* or *disturbed periods*. This latter fact must be true, as we see that the degree of turbulence and irregularity in the ionospheric medium is different in a quiet period or during a storm.

The question is: why is the *variability of dimA with geomagnetic activity* not evident in our results?

The fact is that both in the Solar Maximum year 2001 and Solar Minimum year 2008, geomagnetic storms took place, even though in a different number, but here the time series $y_{2001}\left(t\right)$ and $y_{2008}\left(t\right)$ are simply one-year long time series, *adding together the quiet and stormy periods*. We should then intend $D_{2}=\langle \dim A\rangle$, i.e., *a one-year average* of the time-local dimension of A, and most likely the total amount of stormy time in 2001 to be negligible, to this extent, almost the same as the amount in 2008. To check whether and how much the attractor characteristic dimA changes from the geomagnetically stormy to the quiet dynamics, one should make an analysis *separately for stormy and for quiet times*, including shorter time series that contain severe geomagnetic activity periods. More time-local analyses will be performed in our future work, focusing on the possible differences due to day-time and night-time conditions.

The calculation of $K_{2}$, with its inverse values (Equation 15), points towards a slightly better predictability of vTEC during the Solar Minimum than during the Solar Maximum (the predictability time horizon is 1 min longer, namely $8.49^{\prime }$ versus $5.32^{\prime }$): this indicates that the Solar Wind impacts on the geospace are able, during the Solar Maximum, to render a more irregular, and hence slightly less predictable, local ionospheric medium.

For the forecast time horizons (Equation 15) applied to the vTEC evolution, an observation has arisen at the end of Section 3, i.e., regarding the interplay between the forecast horizon of a few minutes and the apparent presence, in the vTEC time series, of a quasi-periodicity of about 24 h (not to mention the seasonal quasi-periodicities). Looking at the vTEC dependence on local time, as shown in Figure 7 in some detail, clear 24 h recursive patterns show up. Recursivity (quasi-periodicity) introduces some predictability: one can expect the vTEC to be growing the next morning until the local noon, while in the local afternoon an approximate decrease will take place, and this may be stated with great confidence. Yet, the results (Equation 15) state that, analysing what is precisely the behaviour of *y* around a certain time *t*, the dynamics allows one to infer what $y\left(t+\delta \right)$ will be at most for $\delta \leq 8.49^{\prime }$ for the year of Solar Minimum, and $\delta \leq 5.32^{\prime }$ for the year of Solar Maximum, which is much less than the 24 h period of the repetitive patterns. The questions are, then, why does this kind of recursive pattern not appear in our calculated $K_{2}^{-1}$, and how can one recognize the amount of predictability that the recursive patterns introduce, if any. The point is that the whole construction of $D_{2}$ and $K_{2}$ includes the full time series, with all its time-local different conditions, and one uses these data to give some very local information in V, as the Hausdorff dimension dimA. The predictability horizon is calculated in the same way, i.e., by making all the values $\left\{y\left(t\right)\right\}$ participate in the calculation of a $K_{2}$ value that will take into account all the history of the vTEC analysed. The inputs of such an analysis are all the values of *y* at every *t* along the year at hand, and all the possible single time-scale components composing the vTEC variability (as a practical example of vTEC scale decomposition, see the empirical orthogonal functions used in Ref. [24]): looking at the plot in Figure 7, one may indeed distinguish that the local time dependence of the vTEC shows a rather smooth “background” 24 h quasi-periodic behaviour, on which smaller amplitude fluctuations, taking place on shorter timescales, are superimposed (these fluctuations are not shown in Figure 7, in which only the 24 h-quasi-periodic “trend” is reported). The quasi-periodicity, induced by the insolation driver, is clearly influencing the daily scale component of $y\left(t\right)$. Let us call it $y^{\left(24\;h\right)}\left(t\right)$: one could guess that this $y^{\left(24\;h\right)}\left(t\right)$ is a much more predictable evolution than the whole $y\left(t\right)$, with some much smaller $K_{2}^{\left(24\;h\right)}$, and longer forecast time horizon. The guess at this point is that indeed the predictability due to the quasi-periodic patterns in $y\left(t\right)$ appears *only when one analyses the components $y^{\left(\ell \right)}\left(t\right)$ relative to the time scale ℓ*, comparable with the period of those patterns. This guess makes the authors plan to repeat the analysis, performing some time-scale analysis (e.g., via an empirical mode decomposition) of the vTEC $y\left(t\right)=\sum_{\ell }y^{\left(\ell \right)}\left(t\right)$, to see how the single component $K_{2}^{\left(\ell \right)}$, and $D_{2}^{\left(\ell \right)}$ as well, varies with the scale *ℓ*. Such an analysis will be performed in future works.

As a final point, we want to stress that all the work in this paper pertains to the mid-latitude location of Matera, during one year of Solar Maximum and one of Solar Minimum. In order to understand which physical agents determine the complexity and (un)predictability of the local ionosphere, it will be necessary to extend this investigation to different locations (different magnetic latitudes and longitudes) and different magnetic activity periods, always taking care to distinguish the different time scales. Such future studies would be worth comparing with the results obtained for example by Ref. [8] and Ref. [9], who investigated at middle and low latitudes, respectively, the deterministic chaos present in shorter periods of TEC using Lyapunov exponents and surrogate tests, finding higher values for periods of geomagnetically disturbed conditions than for the quiet ones.

Finally, a very ambitious theoretical issue is to explicitly write the *m*-dimensional system (Equation 2) for the local ionosphere: a much more physical and mathematical effort, and a detailed study of the very large series of data available, will be necessary for this purpose.

## Figures and Tables

**Figure 1 entropy-25-00368-f001:**
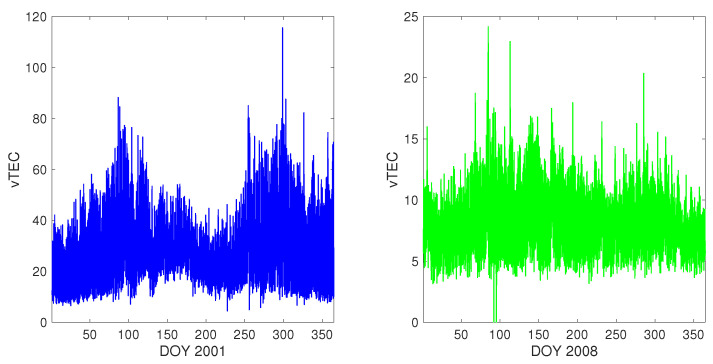
The two plotted vTEC time series $y_{2001}\left(t\right)$ (on the left hand side) and $y_{2008}\left(t\right)$ (on the right hand side). Note that the vTEC scales are different in the two plots. See the text for details.

**Figure 2 entropy-25-00368-f002:**
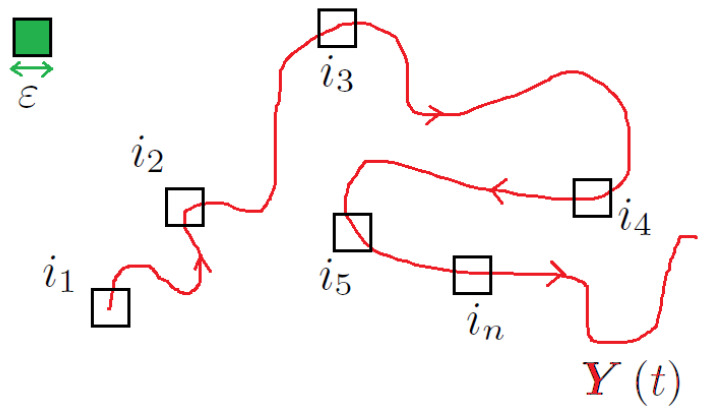
Sketching the ε-coarsening of the points of the trajectory, in order to compute the joint probability distributions $P\left(Y_{1},\ldots ,Y_{n}\right)$. See the text for details.

**Figure 3 entropy-25-00368-f003:**
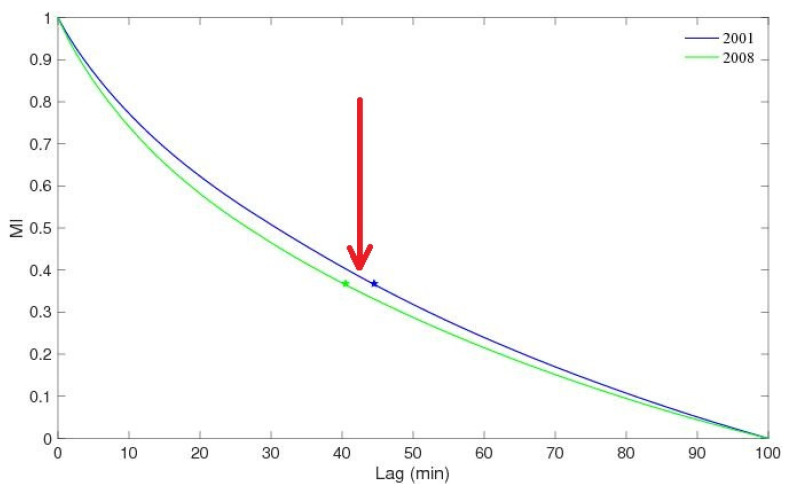
The mutual information (MI) between $T_{\mathrm{vert}}^{\left(\varphi_{0},\lambda_{0}\right)}\left(t\right)$ and $T_{\mathrm{vert}}^{\left(\varphi_{0},\lambda_{0}\right)}\left(t\pm \Delta \right)$ as a function of Δ. The arrow highlights the $\Delta ≃40^{\prime }$ in the correspondence of MI≃0.4bit: this value is strictly correct for the vTEC of year 2008, while it would be too short for that of the year 2001.

**Figure 4 entropy-25-00368-f004:**
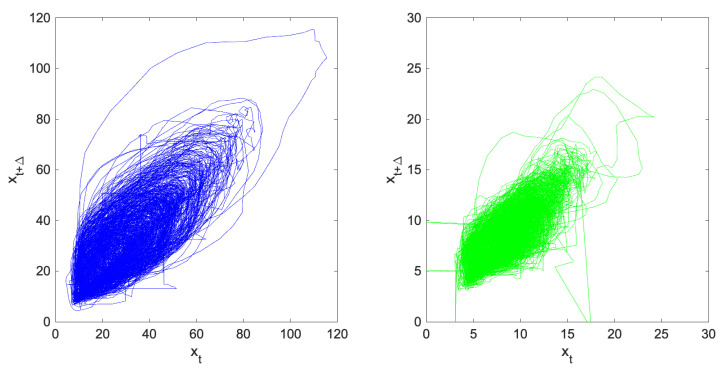
The plots of $y\left(t+\Delta \right)$ against $y\left(t\right)$ for the two years of 2001 (on the left hand side) and 2008 (on the right hand side); see the text for details. Note that the scales of the axis are not the same for the two plots.

**Figure 5 entropy-25-00368-f005:**
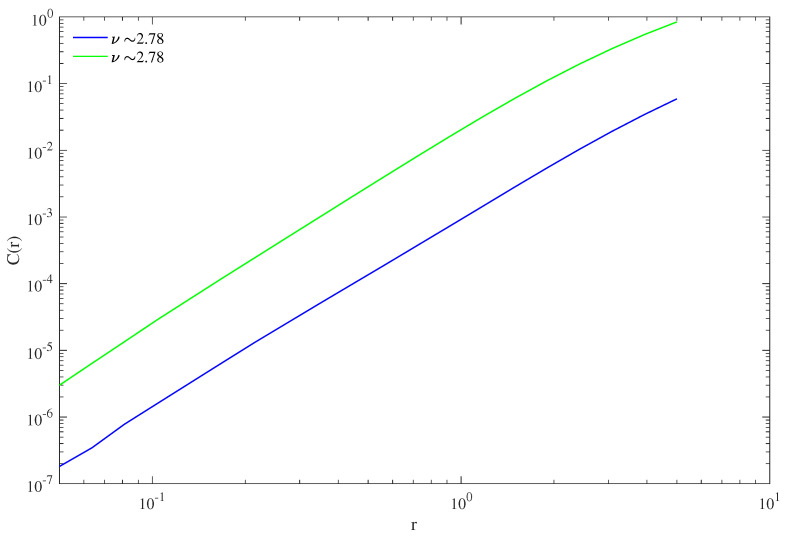
The correlation integral, calculated according to the Grassberger-Procaccia method, for the two time series $y_{2001}\left(t\right)$ in blue and $y_{2008}\left(t\right)$ in green, and plotted against *r* in log-log scale; see the text for details. This determines the value of $D_{2}$ for the two embedded evolutions.

**Figure 6 entropy-25-00368-f006:**
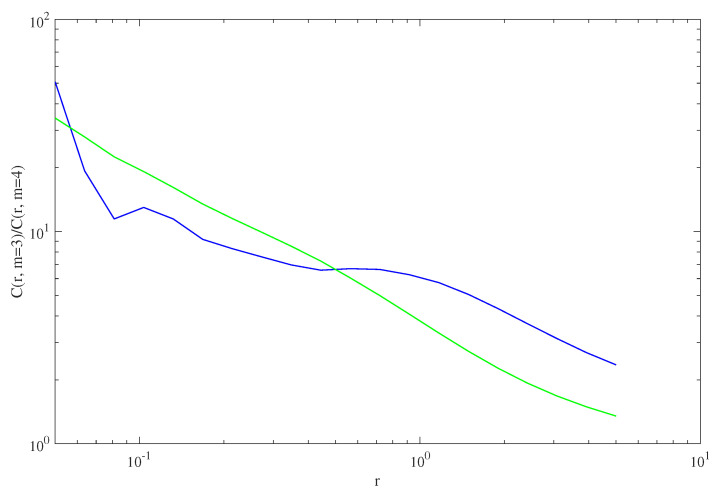
The ratio $\frac{C_{3}\left(r\right)}{C_{4}\left(r\right)}$*Grassberger–Procaccia* calculated for the two time series $y_{2001}\left(t\right)$ in blue and $y_{2008}\left(t\right)$, in green and plotted against *r* in log-log scale; see the text for details. This determines the value of $K_{2}^{-1}$ for the two embedded evolutions, considering that $\frac{C_{3}^{\left(2001\right)}\left(0\right)}{C_{4}^{\left(2001\right)}\left(0\right)}≃50$ and $\frac{C_{3}^{\left(2008\right)}\left(0\right)}{C_{4}^{\left(2008\right)}\left(0\right)}≃35$.

**Figure 7 entropy-25-00368-f007:**
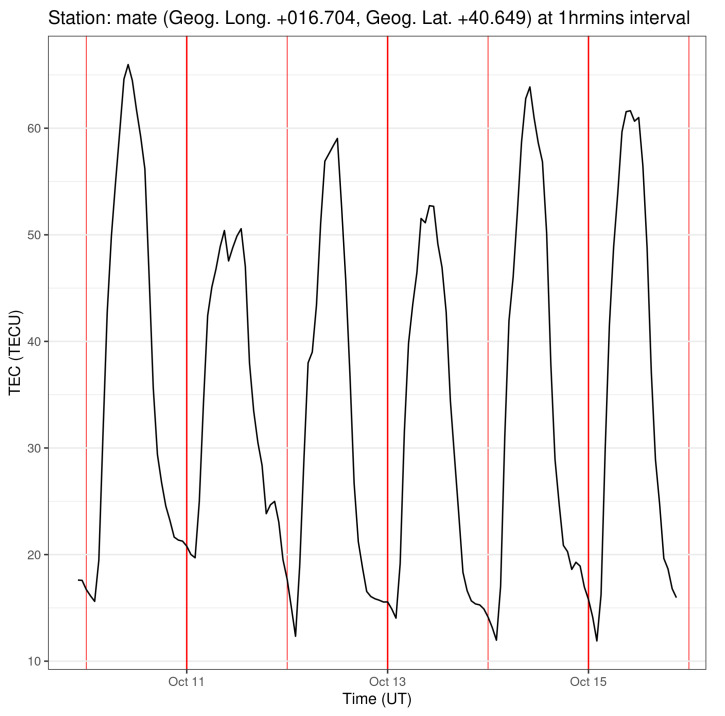
A low resolution version of six days of the time series $y_{2001}\left(t\right)$, in which the quasi-periodicity of the vTEC evolution, due to the sunlight driver, may be appreciated. See the text for details.

## Data Availability

The vTEC data used to perform this study has been obtained from the “ICTP GNSS Calibration Service” freely available at: https://arplsrv.ictp.it//, accessed on 12 February 2023.

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
