# Peer review of "Chaos and Predictability in Ionospheric Time Series"

_entropy, 2023, doi:10.3390/e25020368_

Round 1

Reviewer 1 Report

This paper presents analyses of time-series data of Earth's ionosphere to examine the degree of chaos or predictability of the ionospheric dynamics. Two standard methods of dynamical systems analyses, namely the Hausdolf dimension and Kolmogorov entropy, are used. Although the methods are well known, the analyses are carefully performed and the results showing similarity between the data obtained in the Solar maximum and minimum periods are interesting. The manuscript is well written. One thing I want to make clear is the values in (15). If I understand correctly, K2(2001) = (log_2 50)/30 = 0.188 [bit/sec], which seems different from the inverse of 7.64[min]. Other than this minor comment, I recommend this paper for publication in its present form.

Author Response

The Referee's comment has been gentle and kind. Her/his notation on the values of K2 was definitely important, and allowed us to correct the numbers in Figure 6 as suggested (and as it was necessary).

We sincerely thank the Referee very much.

Reviewer 2 Report

Comments included in the attached file

Author Response

Answers to Reviewer #2

We thank the reviewer for the detailed introduction and comments given to the specific observations and also for his/her very kind compliments that commits us to continuing our research.

Answers to the reviewer observations are written in blue below.

Observation 1 I suppose the data used for modelling is based on GNSS receivers that measure TEC along the signal path. As the ionosphere introduces a time delay in the transionospheric radio-signals, it would be of interest if the authors could specify how was this time delay accounted in the data. This issue also relates to the first reference I have suggested for inclusion. Do the authors collaborate with the IONORING network? Which

directions of investigation are taken into consideration for the future?

We apologize for not having clarified the source of TEC data utilised in the paper. TEC data are taken from the "ICTP Calibrated GNSS TEC Service", web site: https://arplsrv.ictp.it/, that makes use of the TEC calibration technique described in [Ciraolo et al. 2007: [X] Ciraolo, L., F. Azpilicueta, C. Brunini, A. Meza, and S.M. Radicella. Calibration errors on experimental slant total electron content (TEC) determined with GPS. J. Geod., 81, 111–120,2007, DOI: 10.1007/s00190-006-0093-1.]. We thank the observation made by the reviewer and added this information in the revised version of the manuscript (in lines 217-219 of Section 3).

Obs. 2 It will be very interesting to further delve into the subject and describe the attractor characteristic dimension evolution for periods of geomagnetic storms (solar flares) and periods of tranquillity. But this would represent an expected and welcome follow up of the current paper

In section 4 (Conclusions) of the manuscript several lines of research needing further efforts are indicated and we are working in that direction. In particular the importance of investigating in depth the possible influence of geomagnetic activity on the results, as indicated by the reviewer, is mentioned in lines 328-338 of the reviewed manuscript.

Obs. 3 It is stated that “Solar Wind impacts on the geospace are able, during the Solar Maximum, to render more irregular, hence slightly less predictable, the local ionospheric medium”. Again, and this issue should be include on a to do list for the future, there is considerable interest in describing what happens with the ionospheric systems during extreme Space weather phenomena, and I would try to use existing data gathered by the van Allen Probes mission and try to model the behaviour of the ionosphere ( there would be a special interest on the periods when a 3rd van Allen radiation belt emerges after extreme Solar events)

As mentioned in the previous Observation 2, a series of lines of future research necessary to further extend and clarify the results obtained in the present study are indicated in the Conclusions. In our future research endeavor we will also take into account the line suggested by the reviewer. To do that, periods covering extreme geomagnetic activity and new datasets will be considered to investigate its chaotic behavior.

Obs. 4. With respect to the daily and night structure of the ionosphere, and its embedded layers (D layer, E (ES) layer, F layers), it would be again interesting to discriminate between daily and nightly response. Again, this represents an issue to be investigated in the future

The importance of investigating the possible different chaotic behavior during day-time and night-time will be definitely considered in our next study and have been added in the manuscript in lines 331-332, we thank the reviewer for pointing it out. 

Reviewer 3 Report

The manuscript is devoted to a very important topic - understanding variability in the geospace, specifically in the ionosphere. The proposed approach is very sound and promising. The authors are well known specialists on the subject. However, the presentation is not of sufficient quality. Also the text should be more readable for a potential user-geophysist.

1. Order of figures should be as they are mentioned. Logically Fig 3 should be Fig 1.

2 The data set should be explained in more detail. What is the time cadence of the measurements? It is reasonable not only to show one-year period, but also show shorter periods with diurnal periodicity.

3 Sections are very lengthy as well as some paragraphs. The message will be more comprehendable is they will be divided in logical parts.

4. Sometimes explanations of some notions, values or variables are missing. For example, what  is THETA in eq 8 - a step function? "diffeomorphic"?

5 Though theory of the approach is explained in detail, many practical aspects (how these abstract formulas are applied to specific data) are missing. In particular I consider it necessary to add an additional part with specific formulas starting from initial data. In the current form, for a potential user-geophysicist, the message is almost missing.

Author Response

The Third Reviewer’s observations were strong and useful stimuli to improve the presentation of our results, we do thank her/him very much for her/his contribution.

Here please find our replies to Third Reviewer observations.

  1. Figures 1 and 3 have been inter-changed, indeed, as the third Figure is the presentation of the vTEC time series “at a glance”, that may well be the first picture shown.
  2. The time resolution of the time series was already indicated in the original manuscript, yet maybe a little bit en passant. We have turned the sentence into a more explicit one, stressing more clearly how the sampling time is 30 s. As far as the issue of the diurnal periodicity is concerned, the discussion has been slightly enriched, and it has been stated that a new research is already on its way to disentangle periodicity and predictability via multi-scale analysis. Equally, in forthcoming works the Authors are exploring what happens to the analysis when shorter periods of data, with homogeneous geomagnetic conditions, are considered.
  3. Sub-sections have been introduced to have a more step-by-step and focussed distribution of the material presented, both in the general definition discussion and in the presentation of concrete results on the vTEC time series.
  4. Heavyside step function was mentioned explicitly, and the term “diffeomorphic” has been explained in few words (two things are diffeomorphic if related to each other by and invertible, smooth transformation).
  5. Instead of presenting longer explanations of how to apply the formulae in the text, the references in which this is more thoroughly stated as stressed more explicitly. Crucially, the Authors underline that applying the Grassberger-Procaccia formulae to vTEC time series is really equal to applying them to any time series of values.

Round 2

Reviewer 2 Report

The paper can be published in its current state

Reviewer 3 Report

а